# Persistent Infection with Chicken Anemia Virus in 3-Week-Old Chickens Induced by Inoculation of the Virus by the Natural Route

**DOI:** 10.3390/pathogens8020048

**Published:** 2019-04-12

**Authors:** Suttitas Tongkamsai, Meng-Shiou Lee, Ming-Chu Cheng, Hso-Chi Chaung, Yi-Lun Tsai, Yi-Yang Lien

**Affiliations:** 1Department of Veterinary Medicine, College of Veterinary Medicine, National Pingtung University of Science and Technology, Pingtung 91201, Taiwan; pang_pao8@hotmail.com (S.T.); mccheng@mail.npust.edu.tw (M.-C.C.); hcchaung@mail.npust.edu.tw (H.-C.C.); 2Faculty of Veterinary Medicine, Rajamangala University of Technology Tawan-ok, Chonburi 20110, Thailand; 3School of Chinese Pharmaceutical Sciences and Chinese Medicine Resources, China Medical University, Taichung 40402, Taiwan; leemengshiou@mail.cmu.edu.tw; 4Research Center of Animal Biologics, National Pingtung University of Science and Technology, Pingtung 91201, Taiwan

**Keywords:** chicken anemia virus, antibodies, virus replication

## Abstract

Naturally acquired chicken anemia virus (CAV) infection in chickens frequently occurs from 3 weeks of age onward after maternally derived antibodies have decayed. The oral inoculation of older chickens with CAV was reported to have negative effects on cell-mediated immune function, and pathological changes were identified. To date, there has been no complete illustration of an immunological and persistent infection. To understand the pathogenesis of persistent CAV infection, an immunological study of CAV-infected 3-week-old specific pathogen-free (SPF) chickens was carried out by different routes of inoculation. The weight, packed cell volumes, and organ samples were obtained at 7, 14, 21, and 28 days postinfection (dpi). Here, we compared hematological, immunological, and sequential pathological evaluations and determined the CAV tissue distribution in different organs. Neither a reduction in weight gain nor anemia was detected in either the inoculated or the control group. The immune-pathological changes were investigated by evaluating the body and thymus weight ratio and specific antibody titer. Delayed recovery of the thymus corresponding to a low antibody response was detected in the orally inoculated group. This is different from what was found in chickens intramuscularly infected with the same dose of CAV. The CAV remaining in a wide range of tissues was examined by viral reisolation into cell culture. The absence of the virus in infected tissues was typically found in the intramuscularly inoculated group. These chickens were immediately induced for a protective antibody response. A few viruses replicating in the thymus were found 21 dpi due to the regression in the antibody titer in the orally inoculated group. Our findings support that a natural infection with CAV may lead to the gradual CAV viral replication in the thymus during inadequate antibody production. The results clearly confirmed that virus-specific antibodies were essential for viral clearance. Under CIA-risk circumstances, administration of the CAV vaccine is important for achieving a sufficient protective immune response.

## 1. Introduction

Chicken anemia virus (CAV) has recently been classified in the family *Anelloviridae*, genus *Gyrovirus* [1]. The virus causes chicken infectious anemia (CIA) in chickens, which is characterized by aplastic anemia and generalized lymphoid atrophy. The hemocytoblasts in the bone marrow and precursor lymphocytes in the thymus are the major targets for virus infection [2]. Age resistance to clinical infection has been reported after 2 weeks of age. Susceptibility to clinical disease decreases rapidly with the ability to produce the viral neutralizing antibody. However, the subclinical infection that occurs in adult chickens is associated with secondary diseases and production losses [3]. In Taiwan, most field CAVs were isolated from affected chickens at an older age. This may be the result of the horizontal infection after maternal antibodies diminished. Toro et al. [4] compared the pathological severity between 1-day-old and 10-week-old chickens with intramuscular CAV inoculation. They demonstrated that 10-week-old chickens showed no anemia and experienced less severe pathological changes than 1-day-old chicks. The natural route of CAV infection in older chickens is likely the fecal-oral route. van Santen et al. [5] compared the clinical parameters between intramuscular and oral inoculation of CAV in 1-day-old chicks. They found that, in orally inoculated chicks, the CAV genome was detected in the thymus later than it was detected in the thymus of intramuscularly inoculated chicks. Notably, there are limited reports available regarding the effects of CAV on older chickens infected by the oral route. However, Smyth et al. [6] found that CAV antigens were associated with pathological changes in older orally inoculated chickens. This evidence suggested that CAV was capable of infecting certain tissues in older chickens. The pathogenesis of CAV infection is highly related to humoral immunity. The correlation between virus tissue distribution and antibody response has not been studied. The aim of the present work was to study the immunopathogenesis of CAV-inoculated 3-week-old SPF chickens. Values were recorded for packed cell volume (PCV), thymus and body weight ratio, antibody titer, and virus tissue distribution.

## 2. Results

### 2.1. Thymus to Body Weight Ratios

To understand the influence of the inoculation route on immunosuppression caused by CAV infection, the thymus to body weight ratio was used. In both inoculated groups, there were no significant differences in the mean body weight of the infected chickens in comparison to uninoculated chickens throughout the experimental period. The mean thymus to body weight ratios of the intramuscularly (i.m.) infected chickens were significantly lower than those of the control chickens at 7 dpi. However, there were no significant differences between the orally infected group and the control group or between the i.m.-infected group and the orally infected group at 7 dpi (Table 1). Significant differences between the infected and control groups in the mean thymus to body weight ratio were observed at 14 and 21 dpi. At 28 dpi, the difference with respect to each group was not significant. The mean thymus to body weight ratio of the orally infected group was significantly (*p* < 0.05) decreased at 7 and 14 dpi. There were no significant differences within the control group or i.m.-infected group throughout the experimental period.

### 2.2. Hematologic Findings

To verify the alterations in hematopoiesis due to CAV infection, the PCVs were examined. The PCVs in the control chickens ranged from 30 to 31%. The PCVs in orally inoculated chickens ranged from 28 to 29%, and the PCVs in intramuscularly inoculated chickens ranged from 28 to 31% (Table 2). There were no significant differences between the i.m.-infected and the control group or between the i.m.-infected and orally infected group throughout the experimental period. Orally infected chickens maintained low percentages until experiment termination and had significantly reduced PCVs (*p* < 0.05) at 21 dpi. However, no anemic birds were detected in either the CAV-inoculated or the control groups.

### 2.3. Serology

The effect of inoculation route on the induction of anti-CAV antibodies in 3-week-old SPF chickens was monitored using ELISA. All of the selected chickens were confirmed to be free of CAV antibody at the time of inoculation. CAV-specific antibody levels increased after 7 dpi in chickens inoculated both by the oral and i.m. routes with the CAV isolate 1705PT (Figure 1). The antibody level increased after 7 dpi in the i.m.-inoculated group, showing a significant difference from the control group, while no significant differences could be detected between the orally inoculated and the control group. The antibody titers in both infected chicken groups were significantly different at 14 dpi and were highly significant at 21 dpi compared with the control group. The antibody titer of the i.m.-infected group increased significantly at 14 dpi (*p* < 0.01) and even more significantly (*p* < 0.001) at 21 dpi compared with that of the orally infected group. The titer of intramuscularly inoculated chickens reached a maximum level at 21 dpi, at which point the titer remained positive for up to 28 dpi. In contrast, the titer in orally inoculated chickens gradually climbed to a positive level only at 21 dpi. At 28 dpi, the titer in i.m.-inoculated chickens significantly (*p* < 0.01) deceased compared with control chickens but was not significantly different when compared between the orally inoculated and the control group. The control group maintained a negative status throughout the experimental period.

### 2.4. Virus Distribution in Tissues

The virus distribution in tissues was detected by reisolation into MSB-1 cells. Virus infection was most widely distributed at 7 dpi and the thymus tissue had high amounts of virus with a maximum titer in both inoculated groups (Table 3). In the orally inoculated group at 14 dpi, the virus distribution in the thymus tissue was constant and related to the virus distribution in liver tissue, which increased (*p* < 0.001) at 7 dpi. Virus recovery was not detected in any tissue of the i.m.-inoculated group after 14 dpi. The virus remained in the thymus tissue up to 28 dpi even in the orally inoculated group.

## 3. Discussion

The clinical CIA disease is generally observed in young chicks that are infected vertically or during the first two weeks of life. Vaccinated breeder hens transfer antibodies to their progeny, which appear to be completely protective against CAV. Maternal immunity usually disappears by approximately 3 weeks of age. A high prevalence of CIA was detected in older chickens of layer and broiler breeds in Taiwan [7]. The detected cases were naturally occurring horizontal infections with CAV, which most likely occurred by exposure to CAV in the environment. The oral route would be a typical route of CAV transmission in older chickens. Infection of adult chickens with CAV generally does not cause clinical disease. The oral administration of CAV is suggested to induce subclinical disease in older chickens [6]. The characteristic clinical signs of the disease in young chicks are well recognized. However, the mechanism of viral infection in older infected chickens has not been elucidated. We investigated the correlation between virus tissue distribution during subclinical stages of CAV infection and antibody response.

The clinical CIA outbreak in broilers was reported in affected flocks with 3.3% lower than average weight than unaffected flocks [8]. Our observation of the nonsignificant difference in mean body weight between each group throughout the experimental period contradicted findings from earlier studies of 1-day-old chicks [5]. The mean body weight in our studies was not in agreement with those studies. These chickens were infected with higher doses of virus than those used for our study. Wani et al. [9] also reported a significant difference in mean body weight between uninfected and infected chickens at 15 and 25 dpi. This discrepancy could be explained by the fact that these chickens were infected at an older age than the chickens in our study. These results conclude that the dose of inoculation and age of inoculated chickens influence growth performance.

The primary cells targeted during the pathogenesis of CAV infection include hematopoietic precursor and thymic precursor cells in the bone marrow and thymus cortex, respectively [10]. In our study, the constant high-level viral distribution in the thymus of the orally infected group was found to be associated with tissue degeneration, as a lower thymus to body weight ratio was observed from 7 to 14 dpi. Such effects were observed by others when the virus was intraocularly inoculated at a clinically susceptible age [11]. This result is in agreement with the fact that CAV induces apoptosis of thymocytes in infected chickens. The peak distribution of CAV in the thymus of the i.m.-inoculated group were observed at 7 dpi and were related to the lowest thymus to body weight ratio at this time point. The fact that at 14 dpi the mean specific antibody titer of the i.m.-inoculated group was positive while the orally inoculated group remained negative suggests that the virus of the i.m.-inoculated group was eliminated faster than that in the orally inoculated group. As a result, the thymus of the i.m.-inoculated group recovered faster than the orally inoculated group.

Our experimental results revealed that the PCVs of both the CAV-inoculated and control groups had a similar range. Moreover, the PCVs of the orally inoculated group were significantly reduced at 21 dpi, but our chickens did not develop anemia. Similar findings have been previously reported in which subclinical infection with CAV is characterized by the absence of anemia [12]. Smyth et al. [6] also reported that very few infected cells were found in the bone marrow. The latter corresponds with the present study that both inoculated groups were infected with CAV by presenting low virus distribution in bone marrow only at 7 dpi, indicating that age-related resistance does not depend on the route of inoculation. Antibody levels above the positive level of this detection technique are commonly considered protective [13]. Considering the results, the virus was eradicated from the bone marrow of both inoculated groups at 14 dpi even when the mean anti-CAV antibody level was negative in the orally inoculated group. These results indicate that CAV-susceptible cells in bone marrow may be reduced in older chickens or that CAV-infected cells might be eliminated simply with a low titer of antibodies.

Recovery from CAV infection is said to be mainly due to induction of the humoral immune response, as B lymphocytes are resistant to CAV infection [2]. These results contributed to our finding that the virus recovery from the bursa of Fabricius tissue was low in both inoculated groups compared with the thymus. In the present study, anti-CAV antibodies in the serum of the infected chickens were observed from 7 dpi onwards. The appearance of antibodies was associated with viral clearance as the viral distribution in the tissues excluded the thymus after 14 dpi in the i.m.-inoculated group and after 21 dpi in the orally inoculated group. Delayed development of CAV-specific antibodies in orally infected chickens compared to chickens infected by the i.m. route coincided with a reduction in virus accumulation. van Santen et al. [5] explained that, in the oral route of inoculation, the virus titer decreased with innate and local immunologic barriers. This was supported by our result that the virus distribution in the bursa of Fabricius of the i.m.-inoculated group were significantly higher than the orally inoculated group. Although an increase in the anti-CAV antibody titer above the positive line was observed in the orally inoculated group, viral distribution in the thymus was still observed after 21 dpi. These findings corroborate results from other authors who observed that the virus can persist in chickens long after flock seroconversion [14]. Such observations indicate that a prolonged antibody titer may be responsible for the complete recovery of the thymus. Perhaps the virus persists in thymic lymphocytes where it cannot be neutralized by the antibody. Continuous dissemination of the virus from the thymus with insufficient antibody response and the release of infected cells from the thymus into blood circulation might be the cause for the increased virus distribution in the liver of the orally inoculated group at 14 dpi. It is also possible that the virus replicated in the secondary tissue [15]. These conclusions require further investigation to confirm their validity.

In conclusion, the present study is the first to demonstrate a relationship between viral tissue distribution and antibody responses during in vivo CAV infection in older chickens. The findings in the present study support those of Smyth et al. [6], who studied older chickens using parenteral routes of inoculation and suggested that the mechanism of the effect is immunosuppression caused by viral damage to the thymus. To extend our findings, we carried out a small study of chickens inoculated by the intramuscular route. Again, we found that the appearance of antibodies in the serum was associated with marked regression of the viral tissue distribution. Further study may be needed to fully elucidate the persistence of the virus even after the appearance of significantly higher antibody levels. As a result, the CAV vaccine for older chickens may be necessary; however, most authorized vaccines are not recommended for use in pullets younger than 6 weeks of age or slaughter broilers younger than 21 days. Advanced vaccine technology is useful for the immunization of CAV-susceptible chickens.

## 4. Materials and Methods

### 4.1. Birds and Viral Strain

Day-old specific pathogen-free (SPF) chicks were kept in a filtered-air, positive-pressure room under strict isolated conditions. High biosecurity measures were taken to avoid cross-infection. Feed and water were provided ad libitum. Blood samples were collected from nine chickens a few days before inoculation. The samples were examined using an indirect enzyme-linked immunosorbent assay (ELISA) for antibodies against CAV. All experimental procedures on animals were carried out according to the institutional animal care and use committees (IACUCs). The Taiwan field isolate of CAV (1705PT isolate; GenBank Accession No. MK386570) originated from 2-week-old native chickens in 2017 with pale combs and wattles and weakness. The virus was isolated from thymus and liver samples in MDCC-MSB1 cells in accordance with generally accepted procedures [16].

### 4.2. Experimental Designs

Three-week-old chickens (n = 27) were divided into three groups. Each group comprised nine chickens. Groups 1 and 2 were inoculated by intramuscular (i.m.) and oral routes, respectively, with 2 × 10^4.5^ TCID_50_ of the CAV 1705PT isolate, which was the dose used in previous studies in our laboratory. The dose was shown to be capable of inducing subclinical disease in 3-week-old chickens. Group 3 was maintained as the uninfected control group. From 0 to 28 dpi at 7-day intervals, blood samples were collected from the remaining birds in each group for PCVs and for use in the antibody test. Two chickens from each group were weighed and subsequently euthanized for necropsy examination. The thymic lobes were dissected from each side of the neck and weighed. The following tissues were collected and stored at −70 °C: the thymus, the liver, the spleen, the bursa of Fabricius, and the femur.

### 4.3. Hematology

Chickens were bled from the wing vein directly into heparinized microhematocrit capillary tubes. The PCVs were determined following centrifugation at 1200× *g* for 5 min. Chickens were considered anemic if their PCV values were <27%.

### 4.4. Serology

Sera were tested by indirect ELISA for specific antibodies against CAV according to the manufacturer’s (Synbiotic cooperation, Kansas City, MO, USA) directions using 50-fold dilutions of each serum sample. According to the kit’s instructions, serum samples with S/P ratios equal to or less than 0.349 were considered negative, while S/P ratios equal to or greater than 0.350 were considered positive.

### 4.5. Virus Reisolation

Two individual organs in each group were homogenized in phosphate-buffered saline (10% w/v) containing antibiotics. Flat-bottomed 24-well microtiter plates were seeded with an MDCC-MSB1 cell suspension (2.5 × 10^5^ cells per plate). Ten-fold dilutions (10^0^ to 10^−3^) of samples were prepared, 0.1 mL of each dilution was added, and each dilution was pipetted into three replicate wells. Supernatants of tissue homogenates from uninoculated chickens served as controls. Inoculated MSB1 cells were subcultured every 2–3 days, and CAV replication in the cells was assessed by the presence of cytopathic effect (CPE) and conventional PCR [17] at the endpoint. Fifty percent endpoint log titers (log_10_TCID_50_/mL) were calculated by the Spearman–Karber method [18].

### 4.6. Conventional PCR

DNA was isolated from cells collected from 0.5 mL of each culture and extracted in 200 µL using the DNeasy® blood and tissue kit (Qiagen, Germany), according to kit instructions. Specific primer (VP2 gene), forward primer (5′-CTAAGATCTGCAACTGCGGA-3’), and reverse primer (5′-CCTTGGAAGCGGATAGTCAT-3′) were used for viral genome detection. DNA amplification was carried out in a 25-µL reaction volume with 5 µL of extracted DNA, 12.5 µL of GoTaq® green master mix (Promega, Madison, MI, USA), 0.5 µL of each primer (10 µmol), and 6.5 µL of nuclease free water. The cycling program was performed in a thermocycler (T100 PCR Thermal Cycler: Bio-Rad, Hercules, CA, USA), starting with initial denaturation for 5 min at 95 °C, followed by 35 cycles of denaturation (30 s, 95 °C), annealing (1 min, 47 °C), and extension (1 min, 72 °C). After a final extension for 10 min at 72 °C, the samples were held at 4 °C until analyzed. The amplified products were visualized in a 1.5% agarose gel stained with ethidium bromide.

### 4.7. Statistical Analysis

The statistical analyses were performed using analysis of variance (GraphPad Prism 5). A Turkey multiple comparison test was used to compare groups with significant differences (*p* < 0.05).

## Figures and Tables

**Figure 1 pathogens-08-00048-f001:**
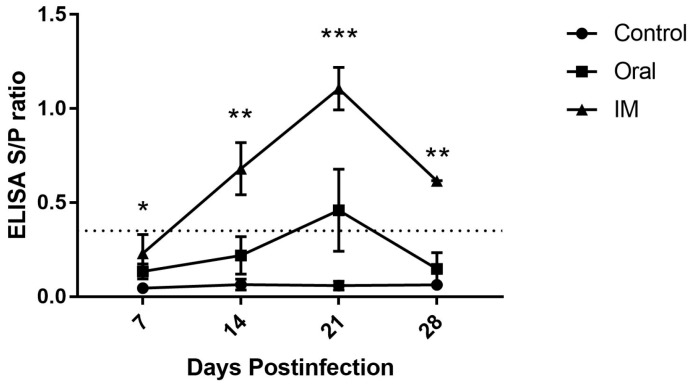
The effect of CAV inoculation on specific antibody levels in 3-week-old infected chickens. Note: The dotted line represents the cut-off titer between negative and positive. The values are represented as the means ± SD; **p* < 0.05, ***p* < 0.01 and ****p* < 0.001.

**Table 1 pathogens-08-00048-t001:** The effect of chicken anemia virus (CAV) inoculation on mean body weights and thymus to body weight ratio.

	Mean Body Weight ± Standard Deviation (g)		Mean Thymus/Body Weight ± Standard Deviation (mg/g)			
dpi	Control	Orally Infected	i.m.-Infected	Control	Orally Infected	i.m.-Infected
7	352.50 ± 7.77 ^aA^	283.75 ± 102.88 ^aB^	349.75 ± 64.70 ^aC^	5.27 ± 0.52 ^eD^	4.64 ± 0.01 ^efE^	4.13 ± 0.52 ^fG^
14	422.50 ± 21.92 ^bA^	411.50 ± 19.09 ^bB^	424.25 ± 28.63 ^bC^	5.44 ± 0.15 ^gD^	3.69 ± 0.01 ^hF^	4.27 ± 0.11 ^iG^
21	523.50 ± 87.68 ^cA^	544.00 ± 82.73 ^cB^	512.50 ± 5.65 ^cC^	5.23 ± 0.34 ^jD^	4.03 ± 0.53 ^kE^	4.35 ± 0.37 ^kG^
28	421.00 ± 127.27 ^dA^	391.50 ± 24.04 ^dB^	385.25 ± 124.80 ^dC^	4.71 ± 0.04 ^lD^	4.20 ± 0.17 ^lE^	4.60 ± 0.10 ^lG^

Values (n = 2) are represented as the mean ± SD, ^A–G^ Among rows; Number with different superscript differ significantly (*p* < 0.05) for a given parameter, and ^a–d^ Among columns; Number with different superscript differ significantly (*p* < 0.05) for each mean body weight values. ^e–l^ Among columns; Number with different superscript differ significantly (*p* < 0.05) for each mean thymus/body weight values.

**Table 2 pathogens-08-00048-t002:** The effect of CAV inoculation on packed cell volumes (PCVs) in 3-week-old infected chickens.

Group.	Days Postinoculation with CAV			
	7	14	21	28
Control	30.44 ± 1.81 ^aA^	31.85 ± 1.34 ^aB^	31.80 ± 1.92 ^aC^	30.00 ± 1.00 ^aE^
Orally Infected	29.66 ± 1.73 ^bA^	29.85 ± 1.34 ^bB^	29.20 ± 1.92 ^bD^	30.33 ± 1.52 ^bE^
i.m.-Infected	29.77 ± 1.40 ^cA^	31.71 ± 1.60 ^cB^	30.00 ± 1.00 ^cCD^	29.66 ± 1.52 ^cE^

The values are represented as the mean ± SD, ^A–E^ Among rows; Number with different superscript differ significantly (*p* < 0.05) for a given parameter and ^a–c^ Among columns; Number with different superscript differ significantly (*p* < 0.05) for a given parameter.

**Table 3 pathogens-08-00048-t003:** The effect of CAV inoculation on the virus distribution in tissue (Log_10_TCID_50_/mL) at 3 weeks old.

	Orally Infected					i.m.-Infected				
dpi	Thymus	Bone Marrow	Liver	Spleen	Bursa of Fabricius	Thymus	Bone Marrow	Liver	Spleen	Bursa of Fabricius
7	4.5 ± 0 ^aA^	2.0 ± 0.5 ^b^	1.5 ± 0 ^cB^	1.5 ± 0 ^bd^	1.5 ± 0 ^b^	4.5 ± 0 ^a^	1.5 ± 0 ^b^	2.5 ± 0 ^bc^	1.5 ± 1 ^bd^	3.83 ± 0.46 ^acd^
14	4.5 ± 0 ^eA^	-	4.5 ± 0 ^eC^	+	-	+	-	-	-	-
21	+	-	-	-	-	-	-	-	-	-
28	+	-	-	-	-	-	-	-	-	-

The values are represented as the mean ± SD, ^A–C^ Among rows; Number with different superscript differ significantly (*p* < 0.05) for a given parameter and ^a–e^ Among columns; Number with different superscript differ significantly (*p* < 0.05) for a given parameter. + = virus present and detected after eighth passage by PCR; - = virus absent after eighth passage.

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
