# Peer review of "Persistent Infection with Chicken Anemia Virus in 3-Week-Old Chickens Induced by Inoculation of the Virus by the Natural Route"

_pathogens, 2019, doi:10.3390/pathogens8020048_

Reviewer 1 Report

Journal: Pathogens - 482085

Title: Persistent Infection with Chicken Anemia Virus in 3-Week-Old Chickens

Induced by Inoculation of the Virus by the Natural Route.

Authors: Suttitas Tongkamsai, Meng-Shiou Lee, Ming-Chu Cheng, Hso-Chi Chaung,

Yi-Lun Tsai *, Yi-Yang Lien *

Comments:

This work is describing a chicken infection with CAV (Chicken anemia virus) in 3-weeks-old birds. Initially, I suggest an English style review.

In the abstract: line 20 – please replace “degenerated” for “metabolized” or “decayed”. Still, the authors must harmonize CAV and CIA in the text. In my opinion, CAV is more suitable.

Results: line 86 – please add the abbreviation of pack cell volumes (PCVs) before the citation in the text. Also, the authors must define, if it is “packed” or “pack”, as mentioned in the table 2 title and the line 86. In the line 125 to 129 – description of “The values are… for a given parameter” – It should be described in Materials and Methods, as well as, the PCR, in detail, for CAV detection that it was mentioned in foot note of table (reference 17). A qPCR for CAV in comparison with the biological assays employed in this study could have been used become it more interesting.

Author Response

Responses to the reviewer comments

Thank the reviewers for the useful comments and suggestions. The authors have performed all the required experiments and addressed each point raised by the reviewers. Also, we have included more information and modified the main manuscript as Track changes.

Reviewer #1:

This work is describing a chicken infection with CAV (Chicken anemia virus) in 3-weeks-old birds. Initially, I suggest an English style review.

Answer: The manuscript was edited for the English style by highly qualified native English speaking editors and was certified by American Journal Experts (AJE). The certificate verification key is BADA-FA79-95C0-BD59-7DF9, the following key could be verified at www.aje.com/certificate.

In the abstract: line 20  please replace degenerated for metabolized or decayed

Answer: As suggested by the reviewer, we have replace degenerated for decayed. Also, we have edited it as Track changes in the revised manuscript.

Still, the authors must harmonize CAV and CIA in the text. In my opinion, CAV is more suitable.

Answer: Actually, CAV and CIA have different meaning, CAV, stand for chicken anemia virus and CIA, stand for chicken infectious anemia. In this manuscript, the term of Chicken anemia virus (CAV)” were used to define the causative virus. For example, in introduction: line 44 CAV has recently been classified..-. However, the term of Chicken infectious anemia (CIA)” were used to define the infectious disease. For example, in discussion: line 131 The clinical CIA disease is generally observed..-.    

Results: line 86  please add the abbreviation of pack cell volumes (PCVs) before the citation in the text. Also, the authors must define, if it is packed or pack, as mentioned in the table 2 title and the line 86.

Answer: As suggested by the reviewer, we have added (PCVs) and also replaced pack for packed. we have edited it as Track changes in the revised manuscript.

In the line 125 to 129  description of The values are for a given parameter It should be described in Materials and Methods, as well as, the PCR, in detail, for CAV detection that it was mentioned in foot note of table (reference 17).

Answer: As suggested by reviewer, we have described the sample preparation in Material and Methods (line 243). As suggested by reviewer, we have described the PCR in 4.6 Conventional PCR (Material and Methods line 251). Also, we have edited it as Track changes in the revised manuscript.

A qPCR for CAV in comparison with the biological assays employed in this study could have been used become it more interesting.

Answer: We agree with the reviewers comment. We also have done the qPCR in this experiment. The results of qPCR were consistent with those of the reisolation method. However, the qPCR may quantify inactivated or non-viable virus particle, therefore the presence of CAV doesnt necessarily mean that the virus is actively, or replicating [1]. Virus reisolation method demonstrates high specificity to detect the complete viral material in several tissues [2] and has used for persistent infection studies [3].

References

1. Smyth, J. A.; Schat, K. A., Virus-induced immunosuppression: chicken infectious anemia. In Immunosuppressive diseases of poultry, Gimeno, I. M., Ed. 2013; pp 91-113.

2. Imai, K.; Mase, M.; Tsukamoto, K.; Hihara, H.; Yuasa, N., Persistent infection with chicken anaemia virus and some effects of highly virulent infectious bursal disease virus infection on its persistency. Res Vet Sci 1999, 67, 233-238.

3. Hoop, R. K., Persistence and vertical transmission of chicken anaemia agent in experimentally infected laying hens. Avian Pathol 1992, 21 (3), 493-501.

Reviewer 2 Report

A good piece of timely research! However I have following query on the manuscript-

1. What does it mean by “persistent” in the article title?

Author Response

Responses to the reviewer comments

Thank the reviewers for the useful comments and suggestions. The authors have performed all the required experiments and addressed each point raised by the reviewers. Also, we have included more information and modified the main manuscript as Track changes.

Reviewer #2:

What does it mean by persistentin the article title?

Answer: Persistent infection occur when the primary viral infection is not cleared by the adaptive immune response. There is many mechanism responsible for establishing a persistent infection. In our work, we assume the virus or viral genome persists in cells of the immune system. Virus material could be diagnosed last for long period.